# Sustained Increase in Very Low Influenza Vaccination Coverage in Residents and Healthcare Workers of Long-Term Care Facilities in Austria after Educational Interventions

**DOI:** 10.3390/vaccines11061066

**Published:** 2023-06-05

**Authors:** Johannes Boyer, Elisabeth König, Herwig Friedl, Christian Pux, Michael Uhlmann, Walter Schippinger, Robert Krause, Ines Zollner-Schwetz

**Affiliations:** 1Division of Infectious Diseases, Department of Internal Medicine, Medical University of Graz, 8036 Graz, Austria; johannes.boyer@medunigraz.at (J.B.); elisabeth.koenig@medunigraz.at (E.K.); robert.krause@medunigraz.at (R.K.); 2Institute of Statistics, Graz University of Technology, 8010 Graz, Austria; 3Geriatric Health Centers of the City of Graz,8020 Graz, Austria

**Keywords:** influenza, vaccine, long-term care facilities, education program

## Abstract

Residents of long-term care facilities (LTCFs) are particularly at risk for influenza infections. We aimed to improve influenza vaccination coverage among residents and healthcare workers (HCWs) in four LTCFs by implementing educational programs and enhanced vaccination services. We compared vaccination coverage before and after the interventions (2017/18 and 2018/19 seasons). Data on vaccination adherence were recorded during a four-year observational period (2019/20 to 2022/23 seasons). Following the interventions, vaccination coverage increased significantly from 5.8% (22/377) to 19.1% (71/371) in residents and from 1.3% (3/234) to 19.7% (46/233) in HCWs (*p* < 0.001). During the observational period (2019/20 to 2022/23 seasons), vaccination coverage remained high in residents but decreased in HCWs. Vaccination adherence was significantly higher in residents and HCWs in LTCF 1 compared to the other three LTCFs. Our study suggests that a bundle of educational interventions and enhanced vaccination services can be an effective method for improving influenza vaccination coverage in LTCFs in both residents and HCWs. However, vaccination rates are still well below the recommended targets and further efforts are needed to increase vaccine coverage in our LTCFs.

## 1. Introduction

Seasonal influenza is a highly contagious viral infection of global importance associated with significant morbidity and mortality each year [1]. According to the World Health Organization (WHO), these annual epidemics are estimated to result in about 3 to 5 million cases of severe illness, and about 290,000 to 650,000 respiratory deaths worldwide [2]. The disease disproportionally affects those with chronic medical conditions and individuals aged 65 years and above [3,4]. Residents of long-term care facilities (LTCFs) are particularly at risk for influenza infections due to close living arrangements and shared caregivers, and they are particularly at risk for severe influenza illness and complications due to frailty and pre-existing medical conditions [4,5,6,7]. A Spanish study found that comorbidities were significantly associated with severity of influenza disease among adults aged 50–79 years [6]. Important underlying diseases in this context include chronic pulmonary or cardiovascular disease, chronic renal disease, hematologic and neurologic disorders, diabetes and obesity [2,6,8]. In addition, patients with comorbidities and of older age experience longer hospitalizations [6].

The role of healthcare workers (HCWs) in the nosocomial transmission of influenza is well documented [9]. A study found that 23% of healthcare workers in acute hospitals had serological evidence of influenza infection during a mild epidemic season [10]. Of the 120 subjects with a seropositive result, about 1/3 could not recall any respiratory infection [10]. This indicates that infected HCWs often continue working while infected, thus participating in nosocomial virus transmission.

Mounting evidence suggests that vaccination against influenza in residents and HCWs in LTCFs may reduce hospital admission, risk of acute cardiovascular events and mortality while also demonstrating cost effectiveness [11,12,13,14]. The WHO and the European Commission have set a target of 75% in all at-risk groups, including those aged over 65 years [15,16]. In spite of national and international recommendations, recent OECD data showed that only 18.3% of the Austrian population aged over 65 years was vaccinated against influenza in 2019 [17,18]. These numbers are among the lowest in Europe [17].

Healthcare personnel are exposed to an increased risk of influenza infection in their work environment. Contact with infected patients and short social distance were associated with an increased likelihood of having a positive test for the influenza virus [19]. An Italian study estimated that over 11,100 working days were lost per year associated with sickness absenteeism during seasonal influenza periods in Italy, resulting in a considerable economic burden [20]. High vaccination coverage among healthcare personnel is crucial to avoid a shortage of workforce [21].

A growing body of literature reports the success of communicative and informative strategies and on-site influenza vaccination in increasing vaccination coverage in HCWs in hospitals [22,23,24,25]. A recent review on campaign strategies to increase influenza vaccine uptake in HCWs included 32 studies [26]. Only two of these had been conducted in nursing homes [26]. A Belgian intervention study conducted on LTCFs found increased vaccination rates from 54% to 68% in the 2015/16 and 2016/17 season, respectively. This was achieved by using a multi-intervention manual focusing on campaign management, education, communication, promotion and easier access to vaccination [27].

The aim of this multicenter study was to evaluate the effect of educational programs and enhanced vaccination services on influenza vaccination coverage among both HCWs and residents of LTCFs in the season immediately following the interventions (2018/2019 season). Our secondary aims were to observe the vaccination coverage in residents and HCWs in the following four influenza seasons.

## 2. Materials and Methods

### 2.1. Setting

We conducted an intervention study at the Geriatric Health Centre of the City of Graz, which is a local institution comprising four LTCFs (total of 388 beds) situated all around the city. Each LTCF is structured in smaller units of 13 to 15 patients sharing a common living room. General practitioners who are located offsite are in charge of medical treatments as well as vaccinations (approximately ten general practitioners per LTCF).

### 2.2. Pre-Intervention: 2017/18 Season

In January 2018, the rates of influenza vaccination of residents and HCWs of 4 LTCFs were determined. The group of HCWs included nursing (auxiliary nurses and nurses), general and administrative staff. In addition, HCWs were asked to respond to a brief online survey anonymously asking for their motivation not to get vaccinated. Residents were asked to fill in a brief survey on paper.

### 2.3. Intervention: 2018/19 Season

In autumn 2018, several interventions took place. HCWs received a personal letter from the infection prevention and control (IPC) team with information on the importance of the vaccination on a personal level as well as for the LTCF. The IPC team held an educational session regarding influenza vaccination in each LTCF together with a physician. During these sessions, characteristics of the disease, modes of transmission, relevance of the disease in the elderly, efficacy of the vaccination and possible side effects of the vaccine were discussed. The influenza vaccination was offered to HCWs free of charge at work during working hours and was administered by the occupational healthcare service of the Geriatric Health Centers of the City of Graz. Buttons with messages such as “Team influenza” or “Vaccination specialist” were handed out after the vaccination. Interventions for residents included an educational session in each LTCF held by the IPC team and a physician on the benefits and risks of the influenza vaccination, posters with LTCF imagery in common rooms and a personal letter for each resident. General practitioners received information on the project and were asked to support the project by encouraging residents and by vaccinating them on site. LTCFs took over the organization and acquisition of the vaccines. Residents had to cover the costs of the vaccine in the 2017/18, 2018/19 and 2019/20 seasons.

In the second week of January 2019, the uptake of influenza vaccination of residents and HCWs in 4 LTCFs was determined (primary outcome).

### 2.4. Observational Period (2019/20 to 2022/23 Seasons)

Semiannual newsletters with information about the LTCF influenza vaccination campaign were sent out to all HCWs by the LTCF administration. HCWs were offered influenza vaccinations free of charge. As of the 2020/21 season, the regional government has offered influenza as well as COVID-19 vaccinations free of charge for all residents in LTCFs. In the second week of January in the years 2020, 2021, 2022 and 2023, the influenza vaccination coverage of residents and HCWs of the 4 LTCFs was determined.

### 2.5. Ethical Approval

The study was approved by the ethics committee of the Medical University of Graz (protocol number 31-103 ex 18/19).

### 2.6. Statistical Analysis

The primary outcome measure was the frequency of influenza vaccinations in the different LTCFs among HCWs and residents during the 2018/19 season. In addition to this primary information, other frequencies from the previous season and from four subsequent seasons were also available, with which the primary frequency was compared. Coding the observation period using a multilevel factor allowed for comparisons with the 2018/19 reference season. We utilized a common-rho beta-binomial logistic regression model that considered the expected frequency of vaccinated persons within a cluster as a function of the explanatory factors observation period, group membership and LTCF. This model allowed for homogenous correlation between all the members within the same cluster and thus accounted for over-dispersion of the variance. Finally, applying a backward elimination technique based on the deviance increase when dropping an effect from the model (equivalent to the application of a likelihood ratio test) yielded all relevant main and pairwise interaction effects at the end.

All *p*-values presented refer to two-sided hypotheses on specific parameters in the final logistic regression model. A test decision was considered significant if the *p*-value was less than 5%.

## 3. Results

In the 2017/18 season, 377 residents and 234 HCWs were included in the study compared to 371 residents and 234 HCWs in the 2018/19 season. Demographic information is shown in Table 1.

### 3.1. Study Period (2017/18 and 2018/19)

In the 2017/18 season, 5.8% (22/377) of the residents and 1.3% (3/234) of the HCWs were vaccinated. Following the interventions, there was a statistically significant increase in vaccination frequencies to 19.1% (71/371) and 19.7% (46/233) in residents and HCWs, respectively, resulting in a respective relative increase of 229.7% and 1422.7%. Table 2 shows the influenza vaccine frequencies among the different LTCFs in the 2017/18 and 2018/19 seasons, respectively. Using a beta-binomial logistic model, we concordantly found that the propensity to be vaccinated was significantly higher in the intervention season (2018/2019) in both residents (*p* < 0.001) and HCWs (*p* < 0.001).

A total of 131 residents and 102 HCWs answered the questionnaire about why they were not vaccinated in the pre-intervention season (multiple reasons could be chosen). Among residents, the most frequently mentioned motive was “I had no opportunity to get vaccinated” (25%) followed by doubts about the effectiveness of the influenza vaccine (16%) and skepticism against vaccines in general (9%). Other reasons included self-perception as being too old, a change in general practitioner and lack of information on influenza vaccination. Among HCWs, the most frequently reported motive was skepticism about the effectiveness of the influenza vaccination (32%). Fifteen percent reported that they were skeptical about vaccinations in general (Figure 1). Eight percent were worried about side effects, whereas three percent were afraid of needles. Nine percent stated the lack of opportunity to get vaccinated against seasonal influenza. Due to high staff turnover (HCWs quitting, changes in leadership) and the resulting lack of comparability, the questionnaire was not repeated in the following seasons.

### 3.2. Observational Period (2019/20 to 2022/23 Seasons)

In the first post-intervention season, 2019/20, the overall vaccination coverage remained stable at 20.3% in residents. In the following seasons, 2020/21, 2021/22 and 2022/23, the coverage increased to 38.5%, 46% and 39.1%, respectively (Figure 2). In the first post-intervention season, 2019/20, the overall vaccination coverage in HCWs remained stable at 24.3%. A peak was reached in the 2020/21 season (44.6%). In 2021/22 and 2022/23, vaccination uptake in HCWs decreased to 16% and 16.7%, respectively (Figure 2).

The propensity to be vaccinated was significantly higher in residents compared to HCWs in the pre-intervention season, 2017/18 (*p* = 0.0378), in 2021/22 (*p* < 0.001) and in 2022/23 (*p* = 0.0017). For both residents and HCWs, the propensity to be vaccinated was significantly higher in the last season, 2022/23, compared to the pre-intervention season, 2017/18 (both for residents and HCWs *p* < 0.001).

The propensity to be vaccinated was significantly higher in residents and HCWs in LTCF 1 compared to the other three LTCFs (vs. LTCF2 *p* < 0.001, vs. LTCF3 *p* < 0.001 and vs. LTCF4 *p* < 0.001).

## 4. Discussion

Our multimodal intervention bundle had a significant impact on vaccination coverage, which increased to about 3-fold in residents and about 15-fold in HCWs in the season following the intervention. For both residents and HCWs, the propensity to be vaccinated was significantly higher in the last season, 2022/23, compared to the pre-intervention season, 2017/18, suggesting a lasting benefit. Several other factors, such as the increased awareness about vaccines in general during the SARS-CoV2 pandemic and the regional government’s free vaccination campaign for residents of LTCFs, may have contributed to the lasting increase.

Our study revealed a very low vaccination rate of 5.8% in residents of LTCFs in the pre-intervention season, 2017/2018. This finding contrasts with the national vaccination recommendations of the Austrian National Vaccination Program that already included chronically ill and immunocompromised patients as well as the elderly in 2017 [28]. The vaccination coverage was also well below the OECD average of 46% and the target of 75% set for the elderly by international authorities [15,16,29]. In other European countries, the reported vaccination coverage among LTCF residents ranged from 46.7% to 93.5% [4]. A European study revealed that the level of health literacy was below average in Austria compared to other European countries such as the Netherlands and Germany [30]. As health literacy has been connected with vaccine acceptance, this may be a possible explanation for our findings [31].

Only 1.3% of HCWs were vaccinated in the pre-intervention season, 2017/2018. In comparison, vaccination coverage in HCWs in other European countries was reported to range from 15.6% to 63.2% (median 30.2%) [4,32]. Vaccination coverage in HCWs is even higher in other geographical areas such as the United States or Australia [4,33,34]. A survey conducted by the Centers for Disease Control and Prevention among HCWs in the United States of America aimed to estimate the influenza vaccination coverage during the 2020/21 season. Overall, 75.9% of HCWs reported receiving an influenza vaccination during the 2020/21 season. By setting, vaccine uptake was highest among HCWs in hospitals (90.1%) and lowest among those employed in nursing homes (66.0%) [35].

A multimodal intervention bundle significantly increased influenza vaccination adherence to 19.1% in residents and 19.7% in HCWs in the season following the intervention. However, the overall vaccination uptake remained modest in both groups in comparison to other European countries. The results of this study are concordant with other investigations showing an increase in influenza vaccination coverage in HCWs through educational programs. In a literature review from 2021 investigating different approaches to increasing vaccination coverage in HCWs, six RCTs with mainly educational intervention were identified with a relative increase in vaccination coverage of 65.9% [26]. One trial was conducted in professional nursing homes in France, demonstrating a significant increase from 27.6% to 33.7% [36]. An intervention study in the United States reported an increase in vaccination rate from 50% to 85% in HCWs in LTCFs following a multimodal intervention [37]. However, in other extensive reviews on interventions to increase influenza vaccination rates among HCWs, educational programs alone had little effect on vaccination rates. Educational programs were successful when combined with other interventions such as enhanced vaccination services, as in our study [26,38,39,40]. While the literature is currently dominated by interventions conducted in hospitals, more publications in other areas with high-risk individuals such as those in LTCFs are becoming available. A systematic review by Bechini et al. specifically addresses interventions for HCWs in LTCFs. While an exact quantification of effects of individual measures was not possible due to the heterogeneity of the studies, trends seem similar compared to studies conducted in hospitals [41]. Regardless of the type of facility, a mandatory vaccination policy seems to be the single most relevant measure to ensure sufficient and sustainable vaccination coverage. The implementation of such strategies led to vaccination coverage exceeding 90% [26,41]. Nevertheless, evaluating other instruments to increase vaccination coverage seems worthwhile, as implementing mandatory vaccine policies may not be feasible in all settings. In fact, most of the studies evaluating mandatory vaccination as a measure to enhance vaccination coverage among HCWs were conducted in North America and may not be successful in most European countries [26,41]. Similarly, an attempt for mandatory influenza vaccination for paramedics in Canada in 2000 resulted in a successful legal challenge against these measures [42]. Additionally, with the increasing relative shortage of paramedics in the western world, institutions have to fear a migration towards other workplaces if the policies are not accepted by the personnel. Last but not least, mandatory vaccination policies cause an ethical dilemma, with the autonomy of the individual on one hand against the moral responsibility to ensure the largest achievable protection for patients on the other hand [43].

During the observational period (2019/20–2022/23 seasons), vaccination coverage among residents remained high, with a peak in the 2021/22 season. Possible reasons are the vaccine campaign by the regional government of Styria offering influenza and COVID-19 vaccines free of charge to all residents of LTCFs starting in 2020/2021 as well as an increased awareness about vaccinations in general due to the COVID-19 pandemic [44,45]. Our findings are in line with data from a systematic review and meta-analysis by Kong et al. The authors showed an increased intention to get vaccinated against influenza in the 2020/21 season compared to the 2019/2020 season. This trend was seen in all geographic areas, with a higher increase in Asia and Europe compared to North America [41]. It should be noted that no study included in this systematic review specifically addressed the population in LTCFs [41].

In HCWs, the peak vaccination adherence was reached in 2020/2021. At this time, vaccines against COVID-19 were still scarce in Austria. One can only speculate that concerns about a surge of influenza happening together with a surge of COVID-19 stimulated many HCWs to get vaccinated in that season. A recent study conducted in a university hospital in Italy also found a significant increase in influenza vaccination coverage during the COVID-19 pandemic 2020/2021 season, with an increase from 61.4% in the 2019/2020 season to 92.9% in the 2020/2021 season in participating HCWs [46].

In the following seasons (2021/2022 and 2022/2023), vaccination coverage dropped in HCWs. This could have been due to the observed high staff turnover (HCWs quitting) and changes in leadership in all LTCFs except LTCF 1. Team leaders remained unchanged in LTCF 1 during the entire study period. They have been important stakeholders in favor of vaccines as part of the IPC strategy throughout the project. The significantly higher propensity of HCWs and residents to be vaccinated in LTCF 1 can most likely be explained by this leadership commitment to the vaccination initiative, as has been described before [37]. The local IPC team has planned to address the new team leaders in the other LTCFs directly to secure their support to increase vaccine coverage in the coming influenza season.

In 2019, the WHO labeled vaccine hesitancy as one of the top ten global health threats because of the risk of reversing progress made in fighting vaccine-preventable diseases. Reasons for vaccine hesitancy are complex and multilayered. In their statement, the WHO identified complacency, inconvenience in accessing vaccines and lack of confidence as key reasons for underlying vaccine hesitancy [47]. A systematic review by Kumar et al. addressing hesitancy for COVID-19 and influenza vaccines in adults found similar results with four main themes, namely, concerns over safety, lack of trust, lack of need for vaccination and cultural reasons [48]. There was a high rate of concern about adverse events [61.4%]. Additionally, a lack of trust in healthcare policies and the pharmaceutical industries and the lack of information resulted in vaccine hesitancy in most studies. Regarding the belief that there is no need for vaccination, about 7% stated that the vaccine would be ineffective, while 30% believed that influenza infection cannot cause death [48]. In our study, we found much lower rates of safety concerns (8%) in both residents and HCWs. On the other hand, the skepticism about the effectiveness of the influenza vaccine in our study exceeded the findings of the systematic review, especially in the HCW group. This may have cultural reasons. In 2017, a survey on vaccine hesitancy in Austria found that a large proportion of participants were concerned about adverse effects (35.9%), doubted the effectiveness of vaccines (35.9%) and expressed distrust towards the pharmaceutical industry (23.1%) [49]. Only about 40% thought themselves to be sufficiently informed about national vaccination recommendations as stated in the Austrian National Vaccination Program [28]. Our results support these findings. The level of skepticism among HCWs about the effectiveness of the influenza vaccine as well as vaccines in general is of particular concern, as our findings suggest an influence of pro-vaccination attitudes in staff on residents.

In the pre-intervention season, one quarter of non-vaccinated residents stated that the lack of opportunity to get vaccinated was their motive. Enhanced vaccination services during the intervention and by the vaccination program of the regional government addressed this particular organizational barrier, probably contributing to a sustained increase in vaccination adherence in residents.

### Strength and Limitations

Our study has several strengths. First, it studies an underrepresented segment of the healthcare system, i.e., nursing homes. Second, it addresses both HCWs and residents. This is particularly important considering the fact that cross transmission can occur in both directions. Third, it includes a four-year observational period to address whether the observed changes were sustainable. This is important as most studies report outcomes only in the short term, while interventions may need a longer time period to prove their effectiveness on the confidence in vaccination [46]. A limitation may be that all four study sites were located in the city of Graz in Austria and were owned by the municipality. This may affect the generalizability of our findings. The multi-intervention approach used in our study resulted in a significant increase in vaccination coverage in both residents and HCWs, but the relative contribution of individual measures cannot be determined.

## 5. Conclusions

A multimodal intervention bundle significantly increased influenza vaccination coverage in the residents and HCWs of four LTCFs in the season of the intervention.

For both residents and HCWs, the propensity to be vaccinated was significantly higher in the last season, 2022/23, compared to the pre-intervention season, 2017/18. Several other factors, such as the increased awareness about vaccines in general during the SARS-CoV2 pandemic and the regional government’s free vaccination campaign for residents of LTCFs, may have contributed to the lasting increase. Overall, vaccination rates are still well below the recommended targets, and further efforts are needed to increase vaccine coverage in LTCFs.

## Figures and Tables

**Figure 1 vaccines-11-01066-f001:**
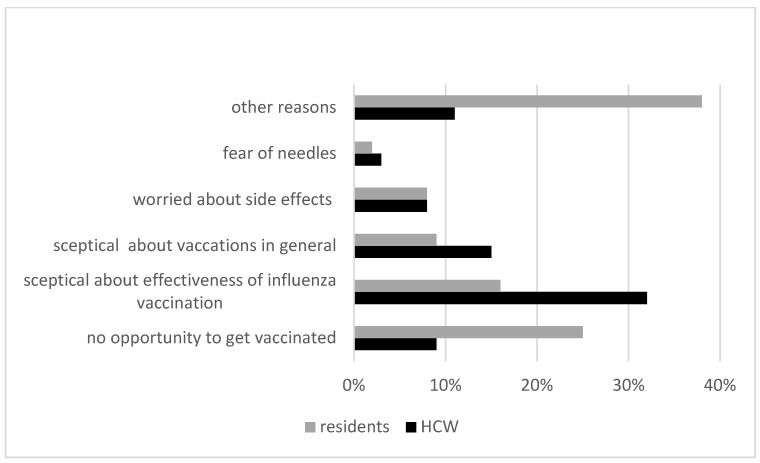
Reasons not to get vaccinated in the pre-intervention season (2017/2018). Multiple reasons could be chosen. Other reasons among residents included: self-perception as being too old, general practitioner changed, not enough information.

**Figure 2 vaccines-11-01066-f002:**
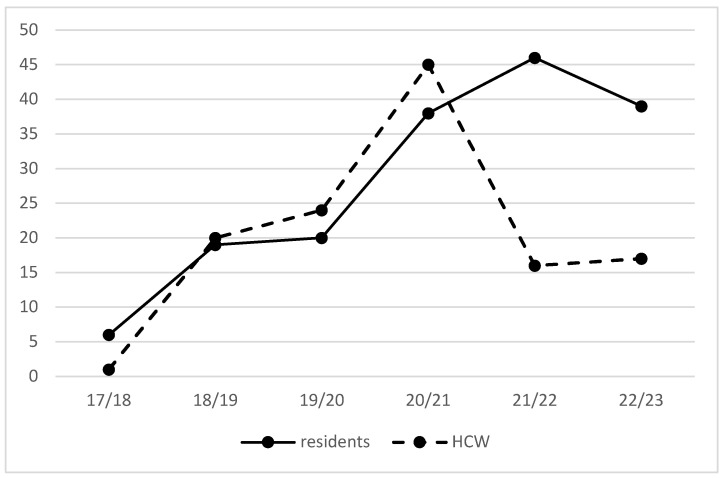
Overall vaccination rates in residents and HCWs from 2017/2018 to 2022/2023. HCWs = healthcare workers.

**Table 1 vaccines-11-01066-t001:** Age and sex distribution among LTCF residents and HCWs.

	Residents (*n* = 377)	HCWs (*n* = 234)
Age in yrs, median, (range)	82 (63–102)	40 (22–58)
Sex		
Male (%)	79 (21.0)	42 (18.0)
Female (%)	298 (79.0)	192 (82.1)

Abbreviations: HCWs = healthcare workers, yrs = years.

**Table 2 vaccines-11-01066-t002:** Vaccination coverage in the 2017/18 (pre-intervention) and 2018/19 (intervention) seasons.

Residents	2017/18 Season Vaccinated/Unvaccinated (%)	2018/19 Season Vaccinated/Unvaccinated (%)
Total	22/377 (5.8)	71/371 (19.1)
LTCF 1	6/96 (6.3)	25/93 (26.9)
LTCF 2	4/94 (4.3)	24/104 (23.1)
LTCF 3	9/90 (10.0)	10/86 (11.6)
LTCF 4	3/97 (3.1)	12/88 (13.6)
**HCWs**	**2017/18 Season** **Vaccinated/Unvaccinated (%)**	**2018/19 Season** **Vaccinated/Unvaccinated (%)**
Total	3/234 (1.3)	46/233 (19.7)
LTCF 1	1/62 (1.6)	19/55 (34.5)
LTCF 2	0/59 (0.0)	9/66 (13.6)
LTCF 3	1/48 (2.1)	12/52 (23.1)
LTCF 4	1/65 (1.5)	6/60 (10.0)

Abbreviations: LTCFs = long-term care facilities; HCWs = healthcare workers.

## Data Availability

Full data available upon request.

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
