# Peer review of "Sustained Increase in Very Low Influenza Vaccination Coverage in Residents and Healthcare Workers of Long-Term Care Facilities in Austria after Educational Interventions"

_vaccines, 2023, doi:10.3390/vaccines11061066_

Round 1

Reviewer 1 Report

This manuscript deals with "Sustained increase of very low influenza vaccination coverage in residents and health care workers of long-term care facilities in Austria after educational interventions" I suggest a minor correction and require a detailed clarification. A correction should be addressed by the authors as follows: The abstract is not well organized; the sentences are incomplete, and there is no sense of continuity. It would be feasible if you included the significance of the current study in the abstract. A brief description of how the authors selected information from the literature in the databases, as well as what time period they searched for, is missing. The authors should justify and expand the information on the advantages of this work. Authors should specify the main experimental conditions used based on the evidence from the literature. Where they briefly describe the most important data reported in the literature in a homogeneous manner and reinforce the relevance of URECs as novel alternatives. Authors should discuss whether the use of this method  represents a solid alternative to existing works. Please add new studies to your manuscript in the discussion section and bold your study novelties:

Author Response

Reviewer 1

We would like to thank Reviewer 1 for his/her thoughtful comments.

This manuscript deals with "Sustained increase of very low influenza vaccination coverage in residents and health care workers of long-term care facilities in Austria after educational interventions" I suggest a minor correction and require a detailed clarification. A correction should be addressed by the authors as follows:

The abstract is not well organized; the sentences are incomplete, and there is no sense of continuity. It would be feasible if you included the significance of the current study in the abstract.

  • The abstract was rewritten to enhance clarity as suggested by the reviewer.

A brief description of how the authors selected information from the literature in the databases, as well as what time period they searched for, is missing.

  • We used the National Library of Medicine’s search tool using the following search terms: influenza vaccine OR vaccination AND long-term care OR nursing home AND health care workers. We did not pre-identify a specific time period but strived to include both relevant and recent publications.

The authors should justify and expand the information on the advantages of this work.

  • The paragraph on the strength of our study was rewritten for clarification. A paragraph on the impact of our study was added to the discussion.
    We feel our study has several advantages. First, it studies an underrepresented segment of the health care system, i.e. nursing homes. Second, it addresses both health care workers and residents. This is particularly important considering the fact that cross transmission occurs in both directions. Third, it includes a four-year observational period to address whether the observed changes were sustainable.

Authors should specify the main experimental conditions used based on the evidence from the literature. Where they briefly describe the most important data reported in the literature in a homogeneous manner and reinforce the relevance of URECs as novel alternatives. Authors should discuss whether the use of this method represents a solid alternative to existing works.

  • Unfortunately, we are not sure how to address this particular comment.

Please add new studies to your manuscript in the discussion section and bold your study novelties:

  • We have included more recent publication in the discussion as well as the introduction. We feel our study has several strengths. The paragraph on the strength of our study was rewritten for clarification.

Reviewer 2 Report

The introduction is too small. Please consider any other research that analyzed the impact of educational intervention on vaccination coverage among HCWs or patients (some suggestions, otherwise  limited: doi: 10.3390/vaccines10030475 - doi: 10.3390/vaccines8010005)

Among results section, why was not investigated the change in reason for not getting vaccinated before and after the intervention? Why this data was not analyzed also in the following seasons?

The data should be implemented with some other descriptive data or data related to the intervention effectiveness. For instance, the increase in coverage among HCWs during 2019/2020 season could be attributable to the fear of COVID-19 (doi: 10.7416/ai.2023.2568), and the relative decline before COVID-19 to level pre intervention could confirm it. 

In Discussion, the subheadings are not correct (only strenghts and limitations could be accepted), please remove it. 

Moreover several studies of other authors could be included: for instance doi: 10.3390/vaccines8010119 and doi: 10.4161/hv.34415 should be considered similarly to many other experience conducted in the field.

  1. The main question addressed by the research is not clear, the impact of the intervention was not fully demonstrated
  2. The topic could be original for the context (LTCF) but should be better explain in the methods the intervention and in the results the impact of the intervention

  3. If authors will not revise, the manuscript add very limited data to the subject area compared with other published material
  4. Should analyze the reasons for vaccination adherence and refusal among the two different population considered
  5. The conclusions are partially consistent with the evidence and arguments presented
  6. The figures and tables are only descriptive similarly to data reported, not very interesting for readers

Some terms are not used in correct context. For instance in the abstract several time the term "vaccnation frequencies" was repeated.

Vaccination adherence or coverage are the only terms to be used.

Author Response

Reviewer 2

We would like to thank Reviewer 2 for his/her thoughtful comments.

The introduction is too small. Please consider any other research that analyzed the impact of educational intervention on vaccination coverage among HCWs or patients (some suggestions, otherwise  limited: doi: 10.3390/vaccines10030475 - doi: 10.3390/vaccines8010005)

  • The introduction was expanded. It now includes recent relevant publications in the field.

Among results section, why was not investigated the change in reason for not getting vaccinated before and after the intervention? Why this data was not analyzed also in the following seasons?

  • We agree with the reviewer that it would have been very interesting to investigate the reasons for non-vaccination after the intervention. However, there was very high staff turnover in this period. Due to lack of comparability, we did not repeat the questionnaire. We added a sentence on this in results section.

The data should be implemented with some other descriptive data or data related to the intervention effectiveness. For instance, the increase in coverage among HCWs during 2019/2020 season could be attributable to the fear of COVID-19 (doi: 10.7416/ai.2023.2568), and the relative decline before COVID-19 to level pre intervention could confirm it. 

  • We agree with the reviewer. We have included a sentence on this and the valuable reference in the discussion.

In Discussion, the subheadings are not correct (only strenghts and limitations could be accepted), please remove it. 

  • The subheadings were removed as suggested.

Moreover several studies of other authors could be included: for instance doi: 10.3390/vaccines8010119 and doi: 10.4161/hv.34415 should be considered similarly to many other experience conducted in the field.

  • The publications have been added to the introduction and the discussion, respectively.

  1. The main question addressed by the research is not clear, the impact of the intervention was not fully demonstrated
  • The main aim of our multicenter study was to evaluate the effect of educational programs and enhanced vaccination services on influenza vaccination coverage among both HCWs and residents of LTCFs in the season immediately following the interventions (season 2018/2019). Our secondary aims were to observe the vaccination coverage in residents and HCWs in the following four influenza seasons.
  • Our multi-modal intervention bundle had a significant impact on vaccination coverage, which increased from 5.8% to 19.1% in residents and from 1.3% to 19.7% in HCWs in the season following the intervention, resulting in a respective relative increase of 229.7% and 1422.7%.
  1. The topic could be original for the context (LTCF) but should be better explain in the methods the intervention and in the results the impact of the intervention
  • We agree with the reviewer, that studies about improving immunization adherence in LTCFs are scarce. Our study attempts to contribute to knowledge on this topic.
  • The methods section was revised to enhance clarity on interventions set.
  • We have included information about the impact of the intervention in the discussion section.

 3. If authors will not revise, the manuscript add very limited data to the subject area compared with other published material.

  • We revised the manuscript as suggested by the reviewer.

  4. Should analyze the reasons for vaccination adherence and refusal among the two different population considered

  • Only the reasons for refusing vaccination were analysed. We agree with the reviewer that the reasons for adherence to vaccination would have been of interest.
  1. The conclusions are partially consistent with the evidence and arguments presented
  • The conclusion section as changed to be more conservative.
  1. The figures and tables are only descriptive similarly to data reported, not very interesting for readers
  • We have made an effort to depict the data contained in the manuscript in a clear and easy-to-understand manner.

Comments on the Quality of English Language:
Some terms are not used in correct context. For instance in the abstract several time the term "vaccnation frequencies" was repeated. Vaccination adherence or coverage are the only terms to be used.

  • We changed the terms as suggested throughout the manuscript.

Round 2

Reviewer 2 Report

Dear Authors,

the manuscript after the first round of review is generally improved in its content and quality of presentation of data.

Another check should be conducted through the text to correct minor text editing such as:

...Residents of long- term care facilities (LTCF)... should be (LCTFs)

...The role of health care workers (HCW) in the nosocomial transmission...should be (HCWs)

The quality of english is good and only minor check should be conducted through the text

Author Response

Dear Reviewer,

We would like to thank you for your detailed review of our manuscript.

Another check should be conducted through the text to correct minor text editing such as:

...Residents of long- term care facilities (LTCF)... should be (LCTFs)

  • This was changed throughout the manuscript.

...The role of health care workers (HCW) in the nosocomial transmission...should be (HCWs)

  • This was changed throughout the manuscript.